# Gene Expression Analysis in Postmortem Brains from Individuals Who Died by Suicide: A Systematic Review

**DOI:** 10.3390/brainsci13060906

**Published:** 2023-06-03

**Authors:** Thelma Beatriz González-Castro, Alma Delia Genis-Mendoza, María Lilia López-Narváez, Isela Esther Juárez-Rojop, Miguel Angel Ramos-Méndez, Carlos Alfonso Tovilla-Zárate, Humberto Nicolini

**Affiliations:** 1División Académica Multidisciplinaria de Jalpa de Méndez, Universidad Juárez Autónoma de Tabasco, Jalpa de Méndez 86205, Mexico; thelma.glez.castro@gmail.com; 2Laboratorio de Genómica de Enfermedades Psiquiátricas y Neurodegenerativas, Instituto Nacional de Medicina Genómica, Ciudad de México 14610, Mexico; 3Servicio de Atención Psiquiátrica, Hospital Psiquiátrico Infantil Dr. Juan N. Navarro, Ciudad de México 14080, Mexico; 4División Académica Multidisciplinaria de Comalcalco, Universidad Juárez Autónoma de Tabasco, Comalcalco 86650, Mexico; dralilialonar@yahoo.com.mx; 5División Académica de Ciencias de la Salud, Universidad Juárez Autónoma de Tabasco, Villahermosa 86100, Mexico; iselajuarezrojop@hotmail.com (I.E.J.-R.); angelmarm19966@gmail.com (M.A.R.-M.)

**Keywords:** expression analysis, brains, suicide, systematic review

## Abstract

Around the world, more the 700,000 individuals die by suicide every year. It is necessary to understand the mechanisms associated with suicidal behavior. Recently, an increase in gene expression studies has been in development. Through a systematic review, we aimed to find a candidate gene in gene expression studies on postmortem brains of suicide completers. Databases were systematically searched for published studies. We performed an online search using PubMed, Scopus and Web of Science databases to search studies up until May 2023. The terms included were “gene expression”, “expressed genes”, “microarray”, “qRT–PCR”, “brain samples” and “suicide”. Our systematic review included 59 studies covering the analysis of 1450 brain tissues from individuals who died by suicide. The majority of gene expression profiles were obtained of the prefrontal cortex, anterior cingulate cortex, dorsolateral prefrontal cortex, ventral prefrontal cortex and orbital frontal cortex area. The most studied mRNAs came of genes in glutamate, γ-amino-butyric acid and polyamine systems. mRNAs of genes in the brain-derived neurotrophic factor, tropomyosin-related kinase B (TrkB), HPA axis and chemokine family were also studied. On the other hand, psychiatric comorbidities indicate that suicide by violent death can alter the profile of mRNA expression.

## 1. Introduction

Suicide is a serious global health problem and one of the primary causes of death worldwide [1]. Moreover, it is one of the most devastating outcomes of individuals with psychiatric disorders [2]. For instance, completed suicide is regarded as the deliberate act of killing oneself and succeeding; commonly, individuals who die by suicide go through a series of suicidal ideations and suicide attempts before completion [1]. There are more studies regarding the psychopathology, risk factor profiles, neurobiology, and neurochemistry of suicide completers (SC) than other traits of the suicide spectrum [3]. Additionally, many factors could be associated to exacerbate or suppress the expression of the genes, factors such as a polymorphism, environment such as childhood abuse, and exposition to trauma such as wars in general post-traumatic stress.

Therefore, several studies have aimed to identify potential suicide biomarkers via the examination of postmortem tissue [4]. Molecular markers and processes identified in postmortem designs may reflect a long-standing risk and/or a more proximal precursor of death by suicide [5]. Specifically, brain tissue has complex patterns of neurochemical and neuroplasticity alterations linked to a variety of psychiatric diseases including suicide [6]. With regards to completed suicide, several reports have proposed a potential causal impact of a differential gene expression on this complex psychiatric trait [7,8]. Therefore, examining the gene expression could help us identify functional variants that might play a more direct role in the SB predisposition [9].

Up until today, the impact of these candidate genetic variants and the risk of suicide is not completely understood [10]. Hence, it is important to analyze conceptual frameworks that enhance our understanding of death by suicide as part of the suicide–spectrum behavior, which could guide us towards supporting or performing hypotheses. Our primary aim was to perform a detailed and updated systematic review of gene expression of postmortem studies from brain tissue of individuals who died by suicide.

## 2. Methods

This study was performed using a predetermined protocol in accordance with the Preferred Reporting Items for Systematic Reviews and Meta-Analysis (PRISMA) statement (Appendix A). The registry of the systematic review is CDR42021274922 (by Gonzalez Castro and Carlos Tovilla).

### 2.1. Search Strategy

PubMed, Scopus and Web of Science databases were used to search for relevant studies published up until May 2023. The search terms were the following: (“gene expression” OR “expressed genes” OR “microarray” OR “qRT–PCR” OR “brain samples”) AND (“suicide” OR “suicidal” OR “suicidality”). References in these studies were examined to identify other possible papers that were not indexed in the databases used. (Figure 1 shows the strategy flowchart.) Search results were uploaded into EndNote X9 for a first screening; subsequently, those files were exported to Covidence for a formal screening.

### 2.2. Inclusion and Exclusion Criteria

To be eligible, the studies had to meet the following criteria: (a) full-text articles, (b) case-control designs, (c) evaluated the association between gene expression and completed suicide, (d) included candidate genes related to suicide risk, (e) analyzed the gene expression using a microarray, next-generation sequencing or a quantitative RT-PCR, (f) published in English, (g) published in peer-reviewed journals and (h) cause of death in controls (anything except suicide).

The exclusion criteria were as follows: (a) data available of no use, (b) non-research papers or (c) duplicates.

### 2.3. Data Extraction

Data of each retrieved publication were independently collected in duplicate by two investigators (González-Castro and Tovilla-Zárate) following a standard procedure. Disagreements were solved through discussion until reaching a consensus. The following data were extracted: (a) first author’s name, (b) publication year, (c) country, (d) gene expression candidate, (e) laboratory methods, (f) suicide methods (cases), (g) diagnostics, (h) sample size, (i) mean age, (j) range age, (k) gender proportion in cases and controls, (l) RNA integrity, (m) postmortem interval and (n) pH of brain. These characteristics were gathered in both cases and controls.

### 2.4. Quality Assessment

The Newcastle–Ottawa Scale (NOS) was applied to assess the quality of the eligible articles. NOS involves three perspectives: study group selection, group comparability and whether the exposure or the outcome of interest for a case–control study is listed in the scale. Each study can receive a maximum of nine stars. Furthermore, all studies were critically appraised using the ROBINS-I tool according to the intervention bias, missing data, confounding factors, outcome bias, report, selection and overall risk bias.

## 3. Results

### 3.1. Selection of the Studies with Gene Expression Analysis

We used the Preferred Reporting Items for Systematic Reviews and Meta-Analyses (PRISMA) guidelines for reporting methodology. Our search provided 4585 studies from electronic databases and 8 from other sources. After the first stage of removed records before screening, 1871 were analyzed. Then, detailed screening showed that 59 studies were eligible for qualitative synthesis in the current systematic review [5,7,8,11,12,13,14,15,16,17,18,19,20,21,22,23,24,25,26,27,28,29,30,31,32,33,34,35,36,37,38,39,40,41,42,43,44,45,46,47,48,49,50,51,52,53,54,55,56,57,58,59,60,61,62,63,64,65,66]. The process of the study selection is depicted in Figure 1.

### 3.2. Characteristics of the Studies

The main population analyzed in the included studies was from North America: the United States of America, Canada and Mexico. Other less studied countries were Hungary, Japan, Slovenia, Spain, Singapore, Germany and Sweden. Concerning the regions and structures of the brain analyzed for gene expression in suicide completers, the most frequent were the prefrontal cortex, anterior cingulate cortex, dorsolateral pre-frontal cortex, ventral prefrontal cortex, orbital frontal cortex, hypothalamus, amygdala and hippocampus. The majority of the studies performed or validated the gene expression by quantitative RT-PCR, microarrays or a western blot analysis (Table 1).

On the other hand, when the quality measurements were applied, we could observe that studies differed in methodological standardization (e.g., type of method applied in gene expression, percent of male/female, among others). Nevertheless, the quality level of the studies did not report an important evidence of bias. However, any results should be taken with caution (Figure 2).

### 3.3. Gene Expression Associated with Completed Suicide

The included results reported several pathways that have been implicated as risk factors of suicide; the glutamate pathway and the γ-amino-butyric acid and polyamine systems have been the most studied. One study of brain expression in suicide completers with and without major depression reported global changes in the synaptic transmission of GABAergic (inhibitory) and glutamatergic (excitatory) systems [30]. In another study of Canadian individuals who died by suicide, the polyamine biosynthetic gene expression was analyzed and it was observed that the AMD1 and ARG2 genes correlated with a decreased methylation of specific CpGs in the promoter region of these genes [46] (Figure 3).

Other pathways commonly associated with completed suicide are the neurotrophic factor systems. The brain-derived neurotrophic factor and tropomyosin-related kinase B (TrkB), even a truncated variant (TrkB.T1), have been found in the frontal cortex and Wernicke area of a completed suicide subpopulation [27,34,35]. The HPA axis genes have been investigated as possible candidate genes for suicide behavior markers, including the corticotropin-releasing hormone receptor genes [11], the FKBP5 and glucocorticoid receptor gene [44]. Other genes commonly investigated are those related to glia or astrocyte cell functioning or proliferation; for instance, we found that the mRNA expression of the chemokines CXCL1, CXCL2, CXCL3 and CCL2 was significantly decreased in the prefrontal cortex (PFC) of suicide completers who had depression when compared with non-suicide individuals [7].

Additionally, a high astrocytic CX gene expression has been found in individuals with depression who died by suicide [56].

Finally, some studies addressed the gene expression of serotonergic and noradrenergic pathways, which are thought to be etiologically relevant to suicide risk and other psychiatric disorders [18,25,32,48,49].

### 3.4. Description of Cases Group (Brains of Suicide Completers)

In this review, we evaluated 1450 individuals who died by suicide; the mean age was 39, ranging from 12 to 94 years old. The majority of suicide completers were men (*n* = 1058). Regarding the characteristics of brain tissue, they showed a mean pH of 6.5, and the postmortem interval was around 24 h. RNA integrity was approximately 7. The main characteristics of each study are described in Table 2.

Regarding the suicide method selection, the most common one in females was drug overdose; while in males, hanging was the most frequently selected. Males showed a strong tendency to use more violent suicide methods (i.e., hanging, a gunshot and jumping) (Figure 4).

### 3.5. Description of the Comparison Group (Brains of Non-Suicidal Individuals)

The comparison group consisted of 1314 individuals with a mean age of 43 years; they died by any other causes (Table 3). The majority of individuals in this group were also men (*n* = 972). The brain samples in the comparison group had a mean pH of 7 and a postmortem interval of 23 h.

## 4. Discussion

The use of postmortem brain samples of individuals who died by suicide leads to new opportunities to study molecular mechanisms underlying suicide behavior [67]. Research focusing on the identification of candidate genetic factors could increase our knowledge of different neurobiological elements responsible for this pathology [68]. Hence, in the present work, we aimed to perform a systematic review of the gene expression association studies on postmortem brain samples from completed suicide.

### 4.1. GABAergic/Glutamatergic Systems

Gamma-amino butyric acid (GABA) and glutamate are the major inhibitory and excitatory neurotransmitters in the mammalian central nervous system (CNS), respectively, and they are thereby involved directly or indirectly in several mental aspects, such as learning, memory, cognition and mood regulation, among other normal functions [69]. Some studies have reported an association of the GABAergic/glutamatergic gene expression with suicide behavior. One study found that the expression of both GABA and glutamate genes is increased in the anterior cingulate cortex of individuals with depression who died by suicide when compared with individuals with depression who died by other causes [70]. Specifically, one study found that the subunit of GABA_A_ receptors, GABRG2, had a higher gene expression in the dorsolateral prefrontal cortex of individuals with major depression [16], while another study found lower GABRG2 brain expression in suicide behavior individuals who died by suicide [71]. We have found strong evidence that the mRNA expression of glutamatergic and GABAergic proteins is similarly altered in completed suicide. Then, is there a disturbed balance between neuronal/brain excitation and inhibition in individuals who died by suicide? It is plausible to assume that brains of suicide completers started with variations in neuronal/brain excitation followed by proportional changes in neuronal/brain inhibition [47,66].

### 4.2. Polyamine System

The cellular roles of polyamines (putrescine, spermidine, spermine and agmatine) include the modulation of synaptic activity and ion channels that participate in the excitability of the neuronal network, as well as the regulation of gene transcription and post-transcriptional modifications [72]. One of the most common genes studied of the polyamine system is the spermine/spermidine N1-acetyltransferase (SSAT) gene. In a Canadian population, for instance, a study found a significant downregulation of SSAT1 in suicide completers [73]. A subsequent study hypothesized that the dysregulation of the SSAT1 gene in suicide completers could be influenced by miRNA post-transcriptional regulation. In fact, this group also found that several miRNAs showed a significant up-regulation in suicide completers when compared with non-psychiatric controls [47]. This altered gene expression of SSAT1 would be expected when there is homeostatic disruption, causing increases in spermine levels, spermidine levels, or both. Additionally, an analysis of transcriptomic and DNA methylation profiles of 21 individuals who died by suicide showed that polyamine oxidase (PAOX) gene expression is up-regulated, which provides a better explanation of the altered levels of polyamines in the brain of completed suicide [63]. Together, these findings suggest that changes in polyamine levels may have deep effects, due to the multiple processes that polyamines contribute to in the CNS.

### 4.3. Neurotrophic Factor Systems

Neurotrophic factors, also known as the neurotrophin signaling system, are a wide variety of polypeptides essential for the development and survival of neurons in the central and peripheral nervous systems [74]. The hypothesis of the relation between neurotrophic factors and behavior is widely proposed by some investigators; in suicide behavior, it has been suggested that alterations in gene expression of neurotrophic factors partly underlie changes in plasticity observed in the brains of suicide completers [75]. One of the most important neurotrophins is the brain-derived neurotrophic factor (BDNF); after binding and activating the receptor tyrosine kinase B (TrkB), it is directly involved in the functioning of neurons and the synapsis [76]. For instance, one study reported that mRNA levels of BDNF and TrkB were reduced in the PFC and hippocampus of suicide completers when compared with individuals who died of other causes [13]. Another report indicates that the BDNF promoter is hypermethylated in the brain of suicide completers, which could have contributed to the downregulation of BDNF expression in suicide completers [34]. Overall, the findings suggest that BDNF and TrkB are promising markers of suicide behavior. Gene expression in postmortem studies underly the dysregulation of specific regions of the brain and neuronal plasticity.

### 4.4. HPA Axis Genes

Our research indicates a dysregulation of the hypothalamic–pituitary–adrenal (HPA) axis stress response activity, as well as an impairment of other typical neurotransmitter systems [77], in the diathesis of suicide behavior. In suicide completers, the receptors of the corticotropin-releasing hormone (CRH), specifically CRH-R1 and CRH-R2, were highly expressed in the brain, particularly in the pituitary [11]. A study found decreased gene expression levels of the neuron-specific glucocorticoid receptor in suicide completers with a history of child abuse [26]. Additionally, investigations have found that the gene expression of co-chaperone FK506-binding protein 51 (FKBP5) was significantly reduced in the amygdala of individuals who died by suicide when compared with controls [44]. It is well known that in situations where our body does not have a successful control of stressors, the outcome effects can be negative; therefore, we could assume that alteration of any member of the HPA axis could induce an endocrine response affecting the brain and peripheral tissues [44].

### 4.5. Chemokine Family

Chemokines are small proteins with several implications in neuroendocrine regulation, blood barrier permeability control, pre- and post-synaptic modulation and other essential activities for the normal functioning of the relation between the central nervous system and immune system (Nakagawa & Chiba, 2015; Stuart & Baune, 2014). For that reason, it is not surprising that chemokines such as CXCL1 and CCL2, among others, have been implicated in a number of neurological diseases. In fact, some papers suggest that the disruption of the previously mentioned chemokine functions in neurodevelopmental periods or in later life contribute to the pathophysiology of psychiatric traits (e.g., suicide) (Nakagawa & Chiba, 2015). Namely, CXCL proteins are involved in the inhibition of glutamatergic activity in hippocampal neurons and regulation processes of neuroplasticity (Rogers et al., 2011; Tokac et al., 2016).

### 4.6. Considerations and Limitations

We want to highlight that gene expression is controlled by a variety of factors that should be considered when drawing conclusions. There is evidence that epigenetic changes such as post-translational histone modifications could alter gene expression without altering the DNA sequence [78]. Different diagnoses are another important factor. We recognize that it was difficult to determine the changes in gene expression associated with suicide behavior, particularly when analyzing genes associated with other psychiatric disorders such as depression. However, having more detailed information regarding the expression of candidate genes could impact the understanding of the pathophysiology in major depression and the diathesis for suicide, including differences and similitudes. We recognize that in future studies, it will be necessary to take into consideration gender analyses, analyses of other suicide features and analyses of the suicide method selected. For instance, we observed that men usually complete suicide using a violent method, while for women, the most frequent methods used are non-violent. In this sense, several studies have suggested that the courage to carry out self-harm with the aim of death is partially heritable [79]. In particular, the serotonin transporter gene consists of one short (S) and one long (L) allelic variant; the S allele is associated with lower gene expression and is repeatedly associated with violent suicide methods [80].

To sum up, even with the limitations mentioned above, we consider that the findings obtained in the present review are of great value and will help to focus future research on relevant pathways that will help improve our knowledge of suicide behavior. Postmortem studies provide a deep insight of brain neurobiology and how it changes in suicide.

## 5. Conclusions

Our results show that mRNA expression in postmortem brains of suicide completers could be increased or decreased depending on the area, axis and/or pathway studied. On the other hand, psychiatric comorbidities indicate that suicide by violent death can alter the profile of mRNA expression. Therefore, more studies are necessary to determine the role of mRNA expression profiles in order to understand the molecular changes in brains of suicide completers.

## Figures and Tables

**Figure 1 brainsci-13-00906-f001:**
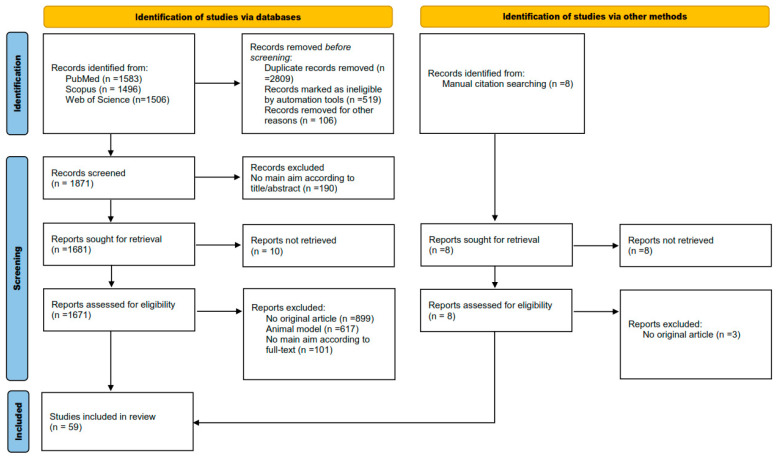
Flow diagram of the search criteria for the systematic review.

**Figure 2 brainsci-13-00906-f002:**
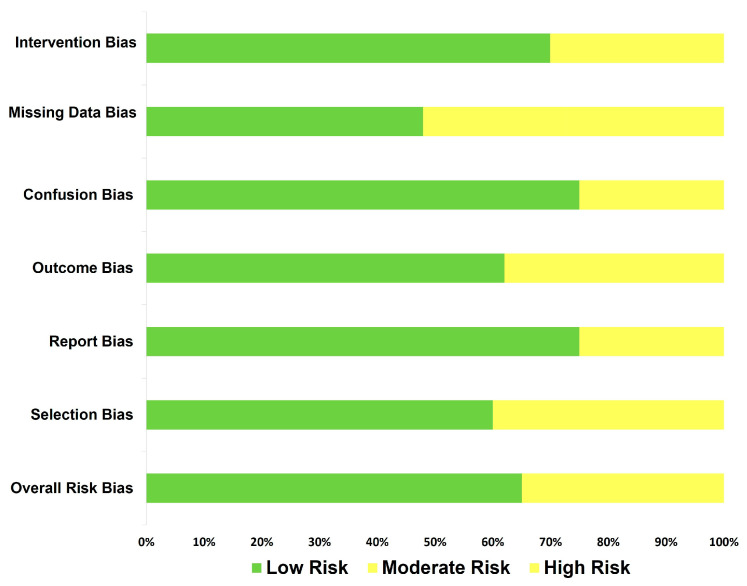
Risk of bias summary: a review of authors’ judgments about risk of bias.

**Figure 3 brainsci-13-00906-f003:**
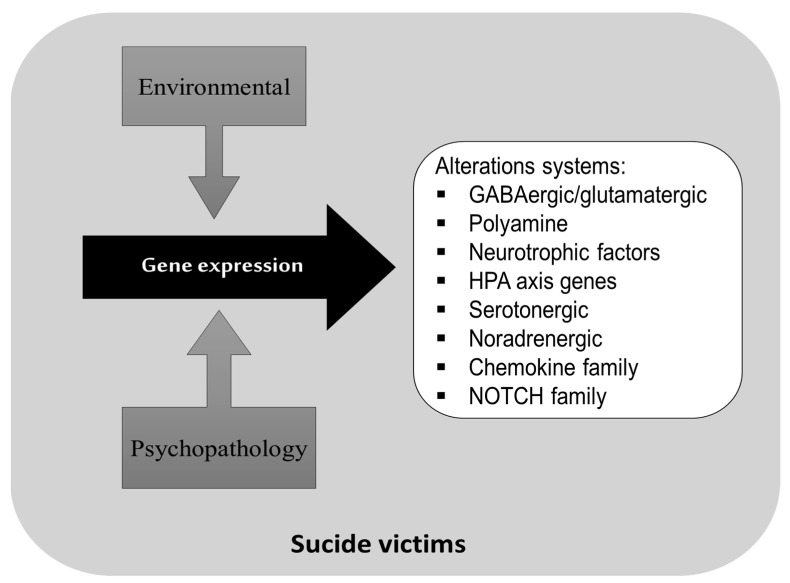
Principal neurobiological systems involved in postmortem studies on brains of suicide completers.

**Figure 4 brainsci-13-00906-f004:**
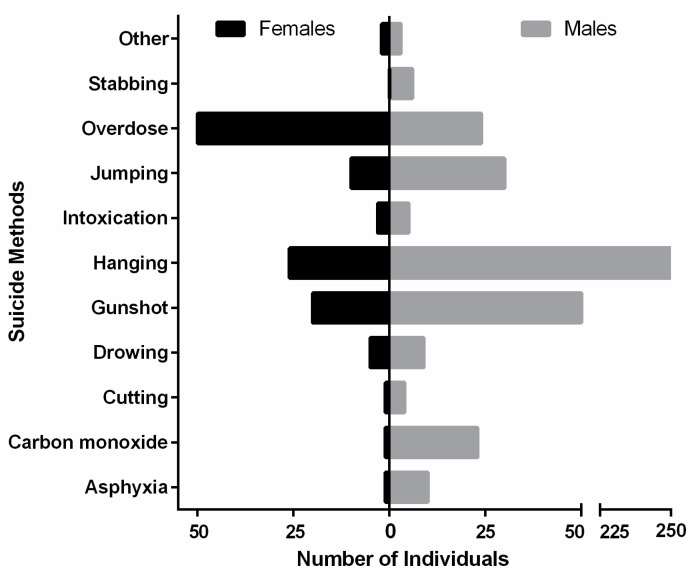
Distribution of suicide method selection by gender.

**Table 1 brainsci-13-00906-t001:** Principal features of studies that searched for an association between gene expression and suicide.

First Author	Year of Publication	Location	Brodmann Area	GE Lab Method	Some Genes Analyzed	NOS
USA						
Hiroi N [11]	2001	AMY, THAL, PG, HPC, CB, SN	NA	qRT–PCR	CRH-R1, CRH-R2	7
Dwivedi Y [12]	2001	PFC, HPC, CB	8, 9, 10	qRT–PCR	MKP2, ERK1/2	8
Dwivedi Y [13]	2003	PFC, HPC	9	qRT–PCR	BNDF, TrkB	9
Sibille E [14]	2004	PFC	9, 47	Microarray	BDNF, TrkB, CREB, HTR1A, HTR2C, ADRA1A, ADRA2B	9
Choudary PV [16]	2005	ACC, DLPFC	24, 9, 46	Microarray	GABAAα1, GABAAß3	8
Kim S [21]	2007	PFC	46/10	Microarray	PLSCR4, EMX2	6
Garbett K [25]	2008	PFC	NA	Microarray	TOB1, NFIA, TLOC, AL119182, HTR2A	9
Pandey GN [28]	2009	PFC, HPC	9	qRT–PCR	GSK-3b	8
Simmons M [33]	2010	DLPFC	NA	qRT–PCR	ADAR1	7
Choi K [36]	2011	PFC, HPC	46	Microarray	CAMK2B, CDK5, MAPK9, PRKCI	7
Sequeira A [40]	2012	DLPFC, ACC, NACC	NA	Microarray	5-HT2A, MT1E, MT1F, MT1G, MT1H, MT1X, MT2A	8
Galfalvy H [42]	2013	DLPFC, ACC	9, 24	Microarray	CYP19A1, MBNL2, KTBBD2, FOXN3, DSC2, CD300LB	8
Ren X [43]	2013	PFC, HPC	9	qRT–PCR	GSK-3b, b-catenin	8
Pandey GN [45]	2013	PFC, HPC, AMY	9	qRT–PCR	GR-α, GR	7
Gray AL [51]	2015	DLPFC	NA	qRT–PCR	GRIN2B, GRIK3, GRM2	9
Fuchsova B [52]	2015	PFC	9	qRT–PCR	GPM6A, GPM6B	9
Gray AL [51]	2015	DLPFC	NA	qRT–PCR	GRIN2B, GRIK3, GRM2	9
Fuchsova B [52]	2015	PFC	9	qRT–PCR	GPM6A, GPM6B	9
Zhao J [53]	2015	DLPFC, ACC	24, 9	qRT–PCR	CRH, NIDD	9
Yin H [54]	2016	DLPFC	9	Microarray	NR3C1	9
Pandey GN [55]	2016	PFC	9	qRT–PCR	SKA2	8
Pantazatos SP [57]	2017	DLPFC	9	NGS	MTRNR2L8, SERPINH1	7
Zhang L [61]	2020	DLPFC, ACC	46, 24	qRT–PCR	P2RY12	8
Zhang Lb [62]	2020	DLPFC, ACC	46, 24	qRT–PCR	TREM2, P2RY12	8
Yoshino Y [5]	2020	DLPFC	9	Microarray	GRP78, GRP94, ATF4C	8
Pandey GN [7]	2021	PFC	9	qRT–PCR	CXCL1, CXCL2, CXCL3, CCL2	8
Canada						
De Luca V [18]	2006	DLPFC	46	qRT–PCR	TPH2	8
Sequeira A [19]	2006	OC, DLPFC, MC	4, 8/9, 11	Microarray	SSAT	7
Sequeira A [20]	2007	AMY, HPC, ACG, PCG	24, 29	Microarray	ADCY8, APLP2	7
Feldcamp LA [24]	2008	DLPFC	46	qRT–PCR	DARPP-32	9
McGowan PO [26]	2009	HPC	NA	qRT–PCR	NR3C1	7
Ernst C [27]	2009	FC, CB	4, 6, 10, 11, 44, 45, 46, 47, 8/9	Microarray	TrkB.T1	6
Klempan TA [29]	2009	OFC, IFG	4, 6, 8/9, 10, 11, 20, 21, 24, 29, 38, 44, 45, 46, 47	Microarray	QKI	9
Sequeira A [30]	2009	ACC	NA	Microarray	GABARAPL1, GABARA4, GABARB1, GRIA3, GRIA4, GRIA1	9
Lalovic A [31]	2010	DLPFC, OFC, VPFC	8/9, 11, 47	Microarray	FADS1, LEPR, PIK3C2A, SCD	8
Fiori LM [37]	2011	AMY, CB, HPC, HPT, NACC, THAL	4, 6, 8/9, 10, 11, 20, 21, 24, 29, 38, 44, 45, 46, 47	Microarray	SAT1, ALDH3A2, AMD1, ARG2	7
Smalheiser N [39]	2012	PFC	9	RT-PCR	DMNT3b, BCL2	9
Labonté B [41]	2013	HPC, DG	NA	Microarray	NR2E1, GRM7	8
Gross J [46]	2013		44	Microarray	OAZ1, OAZ2, AMD1, ARG2	7
Lopez JP [47]	2014	PFC	44	qRT–PCR	SAT1, SMOX	7
Nagy C [56]	2017	MDTHAL, CN, CBTX, CTX	4, 17	qRT–PCR	CX30, CX43	8
Postolache TT [64]	2020	DLPFC, ACC	46, 24, 32, 33	qRT–PCR	CRAMP	9
Squassina A [65]	2020	ACG	24	qRT–PCR	PRKAB2, CREB1, PTEN, PRKAG1, PTPN11, INSR	9
Mexico						
Cabrera B [60]	2019	PFC	9	Microarray	ARL16, KLHL28, SUCLA2, ATP6V0C, TRAK2, CDK19, FNBP1	8
Cabrera-Mendoza Bb [63]	2020	DLPFC	9	Microarray	BBS4, NKX6-2, AXL, CTNND1, MBP, PAOX	8
Romero-Pimentel A [66]	2021	DLPFC	9	Microarray	ADCY9, CRH, NFATC4, ABCC8, HMGA1, KAT2A, EPHA2, TRRAP	9
Asian countries						
Yanagi M [17]	2005	AMY	NA	Microarray	14-3-3 ε	8
Tochigi M [22]	2008	PFC	10	Microarray	CAD, ATP1A3	9
Sherrin T [15]	2004	CB, CG, PFC	NA	qRT–PCR	CCKB	8
European countries						
Thalmeier A [23]	2008	OFC	11	Microarray	AMPH, CDH12, CDH22, CHGB, MYR8, PENK, PTPRR, SCN2B	9
Perroud N [32]	2010	VPFC	11	qRT–PCR	TPH2	7
Keller S [34]	2010	Wernicke	NA	Microarray	BDNF	7
Keller S [35]	2011	Wernicke	8, 9	Microarray	TrkB	7
Zhurov V [38]	2012	FPC	10	Microarray	MEF2D, TFE3, PLAGL1, C1D, XRCC5, EP300, FMR1, VTA1	8
Pérez-Ortiz JM [44]	2013	AMY	NA	qRT–PCR	FKBP5, GR	7
Du L [49]	2014	FPC, OFC	10, 11, 12, 45, 47	qRT–PCR	COMT	8
Monsalve E [50]	2014	DLPFC, AMY	NA	qRT–PCR	NOTCH2, NOTCH1, NOTCH3, NOTCH4, DLL4, JAGGED1	9
García-Gutiérrez MS [58]	2018	DLPFC	9	qRT–PCR	CB2r, GPR55	8
Kouter K [59]	2019	PFC, HPC	9	NGS	NRIP3, ZNF714	8
Mixed populations						
Di Narzo AF [48]	2014	OFC, ACC	11, 25	Microarray	5 HT2CR	7
Cabrera-Mendoza B [8]	2020	DLPFC	9	Microarray	GRM3, GRM8, GRIA2, GRIN2A, GRIN2C	8

PFC: prefrontal cortex; DLPFC: dorsolateral prefrontal cortex; ACC: anterior cingulate cortex; ACG: anterior cingulate gyrus; HPC: hippocampus; HPT: hypothalamus; AMY: amygdala; THAL: thalamus; PG: pituitary; CB: cerebellum; SN: substantia nigra; CG: cingulate gyrus; PCG: posterior cingulate gyrus; OC: orbital cortex; MC: motor cortex; OFC: orbitofrontal cortex; FC: frontal cortex; IFG: inferior frontal gyrus; VPFC: ventral prefrontal cortex; NACC: nucleus accumbens; DG: dentate gyrus; MDTHAL: mediodorsal thalamus; CN: caudate nucleus; CBTX: cerebellar cortex; CTX: cerebral cortex; qRT–PCR: real-time quantitative reverse transcription PCR; NGS: next-generation sequencing; NA: not available. USA: United States of America; NOS: Newcastle-Ottawa Scale.

**Table 2 brainsci-13-00906-t002:** Summary of main characteristics of suicide victims and brain samples of the studies included.

First Author	Diagnostic	N	Mean Age	Range Age	M	F	RIN	pH	PMI
Hiroi N 2001 [11]	-	9	3.8	17–72	3	6	-	-	-
Dwivedi Y 2001 [12]	DD	11	36.2	21–53	5	6	-	-	17.8
Dwivedi Y 2003 [13]	Psychiatric	27	41	21–87	19	8	-	6.1	19.2
Sibille E 2004 [14]	DD	19	44.6	-	14	5	-	-	16.5
Sherrin T 2004 [15]	Psychiatric	10	37.2	16–54	9	1	-	-	16.6
Choudary PV 2005 [16]	DD	9	-	-	7	2	-	-	-
	BD	6	-	-	5	1	-	-	-
Yanagi M 2005 [17]	Psychiatric	14	43.9	26–65	5	9	-	-	18.2
De Luca V 2006 [18]	SCHZ, BD	23	42.6	-	11	12	-	6.47	38.5
Sequeira A 2006 [19]	With DD	16	34	18–53	16	0	-	6.49	22.34
	Without DD	8	35.12	21–51	8	0	-	6.3	24.25
Sequeira A 2007 [20]	Without DD	8	35.1	21–51	8	0	-	6.3	24.3
	With DD	18	36.5	19–53	18	0	-	6.5	24.1
Kim S 2007 [21]	BD	22	44.3	-	11	11	-	6.4	36.8
	SCHZ	10	34.5	-	6	4	-	6.4	35.3
Tochigi M 2008 [22]	DP	11	46	-	6	5	-	-	27
	BD	11	39	-	8	3	-	-	32
	SCHZ	13	44	-	8	5	-	-	33
Thalmeier A 2008 [23]	Psychiatric	11	55.4	33–81	8	3	-	6.72	59.7
Feldcamp LA 2008 [24]	SCHZ	6	36.5	-	3	3	-	6.6	36.8
	BD	16	45.1	-	8	8	-	6.4	39.34
Garbett K 2008 [25]	SCHZ	6	38	25–50	3	3	-	6.9	19.4
McGowan PO 2009 [26]	With CA	12	34.2		12	0		6.3	24.6
	Without CA	12	22.8		12	0		6.5	39.0
Ernst C 2009 [27]	Psychiatric	28	39	18–72	28	0	-	6.5	26
Pandey GN 2009 [28]	Psychiatric	29	16.17	13–20	17	12	-	6.17	18.41
	Psychiatric	27	42.7	22–87	17	10	-	6.12	19.52
Klempan TA 2009 [29]	DD	16	36.5	18–53	16	0	>6	-	24.6
Sequeira A 2009 [30]	Psychiatric	10	34	21–51	10	0	7.14	6.32	29
	DD	16	37	18–53	16	0	7.14	6.55	25
Lalovic A 2010 [31]	With DD	15	34.5	19–53	15	0	6.5	6.6	25
	Without DD	7	32.4	21–51	7	0	6.5	6.4	25.9
Perroud N 2010 [32]	Psychiatric	39	47.36	15–94	26	13	-	6.84	37.38
Simmons M 2010 [33]	DD, BD, SCHZ	15	-	-	9	6	-	6.2	31.2
Keller S 2010 [34]	Psychiatric	44	-	15–79	21	23	-	-	-
Keller S 2011 [35]	Psychiatric	19	-	14–59	10	9	-	6.6	<24
Choi K 2011 [36]	Psychiatric	45	41.7	-	25	20	>7	6.5	32.9
Fiori LM 2011 [37]	DD, BD	29	39.8	-	29	0	-	6.6	27
Zhurov V 2012 [38]	DD	10	52.5	-	10	0	7.4	6.63	5.3
Smalheiser NR 2012 [39]	DD	18	40	19–65	16	2	8.98	6.5	10.7
Sequeira A 2012 [40]	DD, BD	15	44	24–77	12	3	7.8	6.82	24.64
	DD, BD	9	43.3	34–56	6	3	8.03	6.91	23.52
	DD, BD	13	42.72	29–58	10	3	8.1	6.8	24.03
Labonté B 2013 [41]	Psychiatric	13	30.9	-	13	0	6.23	6.6	23.2
Galfalvy H 2013 [42]	DD	18	55.8	-	8	10	>7	-	-
Ren X 2013 [43]	Psychiatric	24	15.92	12–20	14	10	7.2	6.21	19
Pérez-Ortiz JM 2013 [44]	Psychiatric	13	40	18–66	13	0	6.15	-	17
Pandey GN 2013 [45]	Psychiatric	24	15.92	12–20	14	10	7.15	6.21	19
Gross JA 2013 [46]	SCHZ, DD, BD	34	38.6	-	34	0	6.7	6.6	33.9
Lopez JP 2014 [47]	DD	15	37.9	-	15	0	6.4	6.6	29.3
Di Narzo AF 2014 [48]	Psychiatric	22	32.18	16–47	16	6	7.64	6.8	24
	DD	10	47.7	26–72	2	8	8.13	6.61	16.1
Du L 2014 [49]	DD	49	48.91	-	35	14	-	6.57	4.9
Monsalve EM 2014 [50]	Psychiatric	13	40	18–66	13	0	6.15	-	17
Gray AL 2015 [51]	DD	34	41.1	16–83	16	18	8.36	6.3	34.6
Fuchsova B 2015 [52]	DD	25	41.92	22–74	12	13	6.6	7	20.16
Zhao J 2015 [53]	DD	17	40	24–63	10	7	-	6.67	29.6
Yin H 2016 [54]	DD	21	52.1	-	13	8	-	-	-
Pandey GN 2016 [55]	DD	24	38.96	18–74	14	10	>6.6	6.96	18.92
	SCHZ	16	36.31	20–54	13	3	>6.6	6.68	16.69
	Psychiatric	12	40.08	19–87	11	1	>6.6	6.67	22.58
Nagy C 2017 [56]	DD	22	39.7	-	22	0		6.7	17.5
Pantazatos SP 2017 [57]	DD	21	52	-	13	8	6.6	6.4	16.1
García-Gutiérrez MS 2018 [58]	Psychiatric	18	43	18–78	18	0	6.26	-	17
Kouter K 2019 [59]	Psychiatric	9	50.56	-	9	0	-	-	21.33
Cabrera B 2019 [60]	With SUD	23	31.95	-	21	2	>7	-	14.91
	Without SUD	20	32.8	-	12	8	>7	-	15.03
Zhang L 2020 [61]	SCHZ	35	43	19–59	26	9	>7	6.5	31.3
Zhang L 2020b [62]	BD	13	44.5	29–59	7	6	8.7	6.5	39.7
Cabrera-Mendoza B 2020 [63]	DD, BD	48	31	-	38	10	>6	-	15.8
Cabrera-Mendoza B 2020b [63]	DD, PD	21	28.4	-	21	0	>6	-	14.3
Postolache TT 2020 [64]	DD	15	39	-	13	2	7.9	6.6	35
Yoshino Y 2020 [65]	DD	43	50.3	-	26	17	8	6.8	17.9
Squassina A 2020 [66]	BD	7	40.6		4	3			
Pandey GN 2021 [67]	DD	24	38.95	19–74	14	10	≈7	6.95	18.91
Romero-Pimentel AL 2021 [68]	Psychiatric	35	33.11	-	35	0	7.4	-	11.2
Overall		1450	39	12–94	1058	392	7	6.5	24

BD: Bipolar disorder; DD: Depression disorder; SCHZ: Schizophrenia; CA: Child abuse; SUD: Substance-use disorder; N: Sample size; M: Male; F: Female; RIN: RNA integrity numbers; postmortem interval.

**Table 3 brainsci-13-00906-t003:** Summary of main characteristics of controls and brain samples of the studies included.

First Author	Diagnostic	N	Mean Age	Range Age	M	F	RIN	pH	PMI
Hiroi N 2001 [11]	-	7	48.7	-	4	3	-	-	-
Dwivedi Y 2001 [12]	Non-psychiatric	11	37.8	22–46	8	3	-	-	15.7
Dwivedi Y 2003 [13]	Non-psychiatric	21	49.2	22–83	17	4	-	6.1	18.7
Sibille E 2004 [14]	Non-psychiatric	19	44.5		14	5	-	-	18.5
Sherrin T 2004 [15]	Non-psychiatric	10	37.6	20–56	9	1	-	-	18.7
Choudary PV 2005 [16]	Non-psychiatric	7	-	-	6	1	-	-	-
Yanagi M 2005 [17]	Non-psychiatric	14	54.6	28–75	5	9	-	-	12.8
De Luca V 2006 [18]	SCHZ, BD	23	44	-	11	12	-	6.53	32
Sequeira A 2006 [19]	Non-psychiatric	12	35.58	19–55	12	0	-	6.44	25.91
Sequeira A 2007 [20]	Non-psychiatric	13	35.3	19–55	13	0	-	6.5	23.7
Kim S 2007 [21]	BD	23	45.4	-	12	11	-	6.4	36.8
	SCHZ	35	44.3	-	28	7	-	6.4	31.4
Tochigi M 2008 [22]	Non-psychiatric	15	48	-	9	6	-	-	24
Thalmeier A 2008 [23]	Non-psychiatric	10	64.1	48–83	7	3	-	6.71	69.5
Feldcamp LA 2008 [24]	SCHZ	29	44	-	23	6	-	6.5	30.14
	BD	18	46.2	-	9	9	-	6.4	34.3
Garbett K 2008 [25]	Non-psychiatric	6	39	19–52	4	2	-	6.8	18.2
Ernst C 2009 [26]	Non-psychiatric	11	39	28–58	11	0	-	6.5	22
McGowan PO 2009 [27]	Non-psychiatric	12	35.8		12	0		6.5	23.5
Pandey GN 2009 [28]	Non-psychiatric	26	16.46	13–19	18	8	-	6.19	18.41
	Non-psychiatric	20	43.55	22–83	16	4	-	6.1	18.45
Klempan TA 2009 [29]	Non-psychiatric	13	35.3	19–55	13	0	>6	-	23.7
Sequeira A 2009 [30]	Psychiatric	13	35	19–55	13	0	7.14	6.44	24
Lalovic A 2010 [31]	Non-psychiatric	13	37	19–55	13	0	6.5	6.4	22.5
Perroud N 2010 [32]	Mostly non-psychiatric	40	51.13	16–97	27	13	-	6.79	39.58
Simmons M 2010 [33]	Non-psychiatric	15	-	-	9	6	-	6.2	23.7
Keller S 2010 [34]	Non-psychiatric	33	-	13–76	16	17	-	-	-
Keller S 2011 [35]	Non-psychiatric	18	-	13–70	7	11	-	6.8	<24
Choi K 2011 [36]	Psychiatric	38	47.2	-	21	17	>7	6.4	33
Fiori LM 2011 [37]	Non-psychiatric	16	39.8	-	16	0	-	6.6	27
Zhurov V 2012 [38]	Non-psychiatric	9	59.4	-	9	0	6.5	6.5	3.7
Smalheiser NR 2012 [39]	Non-psychiatric	17	35.5	19–63	17	0	9	6.5	26.9
Sequeira A 2012 [40]	Psychiatric	6	59	19–59	4	2	8.5	6.8	25.5
	Psychiatric	6	53.3	44–59	3	3	8	6.76	15.9
	Psychiatric	8	52.3	44–66	6	2	8.1	6.6	27.7
Labonté B 2013 [41]	Non-psychiatric	9	37.4	-	9	0	6.48	6.7	27.8
Galfalvy H 2013 [42]	Non-psychiatric	21	51.8	-	14	7	>7	-	-
Ren X 2013 [43]	Non-psychiatric	24	16.29	13–19	17	7	7.2	6.15	18.13
Pérez-Ortiz JM 2013 [44]	Non-psychiatric	13	46	19–64	13	0	6.71		16
Pandey GN 2013 [45]	Non-psychiatric	24	16.29	13–19	17	7	7.21	6.15	18.13
Gross JA 2013 [46]	Non-psychiatric	34	43.6	-	34	0	6.4	6.5	45.2
Lopez JP 2014 [47]	Non-psychiatric	16	39.8	-	16	0	6.4	6.6	23.8
Di Narzo AF 2014 [48]	Non-psychiatric	29	37.2	19–65	24	5	7.5	6.72	21.4
	DD	24	49.7	16–74	8	16	8.2	6.55	17.7
Du L 2014 [49]	Non-psychiatric	72	64.47	-	46	26	5.96	6.67	4.07
Monsalve EM 2014 [50]	Non-psychiatric	13	46	19–64	13	0	6.71	-	16
Gray AL 2015 [51]	Non-psychiatric	32	39.2	16–65	19	13	8.2	6.5	29
Fuchsova B 2015 [52]	Non-psychiatric	25	42.8	22–72	18	7	6.6	7.01	17.72
Zhao J 2015 [53]	Non-psychiatric	12	47	24–63	8	4	-	6.64	25.3
Yin H 2016 [54]	Mostly non-psychiatric	38	46.9	-	29	9	-	-	-
Pandey GN 2016 [55]	DD	12	49.5	14–74	7	5	>6.6	6.8	17.92
	SCHZ	15	50.6	24–83	10	5	>6.6	6.61	15.33
	Non-psychiatric	24	42.08	19–83	20	4	>6.6	7.02	16.54
Nagy C 2017 [56]	Non-psychiatric	22	41.6	-	22	0	-	6.5	21.5
Pantazatos SP 2017 [57]	Non-psychiatric	29	43.5	-	23	6	7.1	6.5	13.2
García-Gutiérrez MS 2018 [58]	Non-psychiatric	15	46	19–64	15	0	6.35	-	15
Kouter K 2019 [59]	Non-psychiatric	9	53.11	-	9	0	-	-	24.22
Cabrera B 2019 [60]	With SUD	9	30.88	-	8	1	>7	-	17.76
	Without SUD	14	31.78	-	8	6	>7	-	16.84
Zhang L 2020 [61]	Non-psychiatric	34	45	31–60	25	9	>7	6.69	28.5
Zhang L 2020b [62]	Non-psychiatric	34	45	31–60	25	9	8.4	6.69	28.5
Cabrera-Mendoza B 2020 [63]	Non-psychiatric	27	35	-	20	7	>6	-	19
Cabrera-Mendoza B 2020b [63]	Non-psychiatric	6	29.33	-	6	0	-	-	15.53
Postolache TT 2020 [64]	Non-psychiatric	15	36.6	-	15	0	7.9	6.5	34.6
Yoshino Y 2020 [65]	Non-psychiatric	27	48.4	-	16	11	7.9	6.6	18.7
Squassina A 2020 [66]	Non-psychiatric	12	38		3	9			
Pandey GN 2021 [67]	Non-psychiatric	24	42.08	19–62	20	4	≈7	7.01	16.54
Romero-Pimentel AL 2021 [68]	Non-psychiatric	13	32.4	-	13	0	7.2	-	11.6
Overall		1314	43	13–97	972	342	7	6.5	23

SCHZ: Schizophrenia; BD: Bipolar disorder; DD: Depression disorder; SUD: Substance-use disorder; N: Sample size; M: Male; F: Female; RIN: RNA integrity numbers; postmortem interval.

## Data Availability

Not applicable.

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
