# Peer review of "Gene Expression Analysis in Postmortem Brains from Individuals Who Died by Suicide: A Systematic Review"

_brainsci, 2023, doi:10.3390/brainsci13060906_

Round 1

Reviewer 1 Report

This paper reports the results of a systematic review of studies of gene expression in the brains of suicide victims.

The topic addressed by this paper is of considerable clinical and research importance and falls within the scope of the special issue. However, there are several aspects of the manuscript which would benefit from correction or clarification:

1. Abstract: As this paper is a systematic review, the abstract should contain more precise and specific details of review methodology. The results reported in the Abstract should match those included in the manuscript.

2. Introduction:

a. Terms such as "suicide behavior" are non-standard and may cause confusion among other researchers in the field. It is preferable to use standard terms / nomenclature when dealing with suicide-related behavior.

b. It is not clear what the authors mean by "genetic basis of gene expression" (line 51). Gene expression can be influenced by a variety of factors, and post-mortem studies cannot provide significant information on the "genetic basis" of gene expression. The authors should provide a clearer and more succinct rationale for their review.

c. The discussion of genetic, gene x environment and epigenetic factors in suicide, and their relationship with gene expression in brain tissue, is somewhat superficial and does not provide a sufficient background for the current systematic review. The authors should consult recent literature and literature syntheses in this field and rewrite paragraph 1 of the Introduction.

2. Methods:

a. The authors have mentioned registering their review protocol. However, I was unable to retrieve any details using the number provided at either Cochrane, PROSPERO or OSF. I request the authors to kindly provide full details of protocol pre-registration including the site / database where it has been registered, the date of registration, and the registration number.

b. The completed PRISMA checklist may be uploaded as supplementary material along with the manuscript.

c. The search strings (and related details) provided in the manuscript are insufficient. The authors have mentioned only three search terms (gene expression, suicide, and suicidal) which is inadequate for a systematic review of this sort. More technical terms related both to gene expression studies and to suicide should be included. The exact number of citations retrieved for each term (for at least one database) should also be mentioned, to ensure replicability and further extension of these results.

d. The authors report having retrieved a total of 489 citations from three databases. This number is not credible, as even a search of the PubMed database with the terms provided by the authors yields 1476 citations. Please provide database-wise details of the correct number of citations retrieved from each database.

e. Screening of citations usually involves two stages - some citations can be filtered out based on the title and abstract alone, while others require a reading of the full text. This should be mentioned and included in Figure 1.

f. In Figure 1, no explanation for the superscript ** has been provided.

g. Figure 1 refers to 123 citations being retrieved from "registers" - which registers were searched? This should be mentioned in the text as well.

h. The authors refer to "automation tools" being used to mark records as ineligible in Figure 1. Which tool(s) were used?

i. The authors have used the Newcastle-Ottawa Scale (NOS) to assess study quality. Is this scale appropriate for evaluating the quality of gene expression studies, or are there more specific tools available? 

j. What was the rationale for using a cut-off of 6 stars or more on the NOS for including studies in the review? The developers of the tool themselves acknowledge that there is no clear evidence currently available to support such a cut-off.

h. How were confounding factors (e.g., victims' age, gender, or the presence of a medical or psychiatric diagnosis) taken into consideration when appraising the results of included studies?

3. Results:

a. The numbers provided in Section 3.1 do not match those provided in Figure 1; please ensure that these details are consistent throughout the paper.

b. It is not clear what is meant by "most frequent" and "most recurrent" in section 3.2; the two terms are practically synonymous, but different brain regions are listed in each case.

c. Figure 2 does not present the review results in a balanced manner; there is an undue focus on some results (GABA, glutamate, HPA axis, monoamines) while other such as CXCL and NOTCH genes have been ignored.

d. Was any analysis of the relationship between study quality and results attempted? If not, why?

4. Discussion:

a. This section should discuss methodological / quality-related concerns for the included studies, as well as the possible effect of confounders (see 2h. above) and not just repeat the study results.

b. As mentioned in 3c. above, the discussion should focus on all relevant / significant findings and not just the four to five groups of findings that have been repeated most often. Contemporary understanding of suicide has moved beyond simple models based on monoamine / amino acid transmitters to incorporate neural plasticity, stress response, gut-brain axis interactions, neuropeptides, etc. and any positive results pertaining to these (and other related) pathways should also be discussed.

Author Response

REVIEWER 1

This paper reports the results of a systematic review of studies of gene expression in the brains of suicide victims. The topic addressed by this paper is of considerable clinical and research importance and falls within the scope of the special issue. However, there are several aspects of the manuscript which would benefit from correction or clarification:

Comment 1. Abstract: As this paper is a systematic review, the abstract should contain more precise and specific details of review methodology. The results reported in the Abstract should match those included in the manuscript.

Response 1. We thank reviewer #1 for this comment. We have restructured and rewrite the abstract section.

Comment 2. Introduction a. Terms such as "suicide behavior" are non-standard and may cause confusion among other researchers in the field. It is preferable to use standard terms / nomenclature when dealing with suicide-related behavior.

Response 2. We thank the reviewer for this important comment. Accordingly, we have revised the manuscript to standard terms / nomenclature in the manuscript.

Comment 3. Introduction b. It is not clear what the authors mean by "genetic basis of gene expression" (line 51). Gene expression can be influenced by a variety of factors, and post-mortem studies cannot provide significant information on the "genetic basis" of gene expression. The authors should provide a clearer and more succinct rationale for their review.

Response 3. Thanks for your kind comments. We have deleted “genetic basis”

Comment 4. Introduction c. The discussion of genetic, gene x environment and epigenetic factors in suicide, and their relationship with gene expression in brain tissue, is somewhat superficial and does not provide a sufficient background for the current systematic review. The authors should consult recent literature and literature syntheses in this field and rewrite paragraph 1 of the Introduction.

Response 4. We rewrite some information.

Change in the manuscript:

Suicide is a serious global health problem and one of the primary causes of death worldwide [1]. Moreover, it is one of the most devastating outcomes of individuals with psychiatric disorders [2]. For instance, completed suicide is regarded as the deliberate act of killing oneself and succeeding; commonly, individuals who die by suicide go through a series of suicide ideations and suicide attempts before completion [1]. There are more studies regarding the psychopathology, risk factor profiles, neurobiology, and neuro-chemistry of suicide completers (SC) than other traits of the suicide spectrum [3]. Also, many factors could be associated to exacerbate or suppressed the expression of the genes, Factors such as polymorphism, environmental such as childhood abuse, exposition to trauma such as wars in general post-traumatic stress

Comment 5. Methods a. The authors have mentioned registering their review protocol. However, I was unable to retrieve any details using the number provided at either Cochrane, PROSPERO or OSF. I request the authors to kindly provide full details of protocol pre-registration including the site / database where it has been registered, the date of registration, and the registration number.

Response: We have update the register of the systematic review in PROSPERO. This for that all public can retrieve any details of the systematic review register.

Comment 6. Methods b. The completed PRISMA checklist may be uploaded as supplementary material along with the manuscript.

Response: We appreciate the comment and we agree. We have uploaded the PRISMA checklist.

Comment 7. Methods c. The search strings (and related details) provided in the manuscript are insufficient. The authors have mentioned only three search terms (gene expression, suicide, and suicidal) which is inadequate for a systematic review of this sort. More technical terms related both to gene expression studies and to suicide should be included. The exact number of citations retrieved for each term (for at least one database) should also be mentioned, to ensure replicability and further extension of these results.

Response 7.

Change in the manuscript:

2.1. Search strategy

PubMed, Scopus and Web of Science databases were used to search for relevant studies published up to May 2023. The search terms were the following: (“gene expression” OR "expressed genes" OR "microarray" OR "qRT–PCR” OR “brain samples”) AND (“suicide” OR “suicidal” OR “suicidality”).

Comment 8. Methods d. The authors report having retrieved a total of 489 citations from three databases. This number is not credible, as even a search of the PubMed database with the terms provided by the authors yields 1476 citations. Please provide database-wise details of the correct number of citations retrieved from each database.

Response 8.

Change in the manuscript:

3.1. Selection of the studies with gene expression analysis

Our search provided 4,585 studies from electronic databases and 8 from other sources. After the first stage of removed records before screening 1,871 were analyzed.

Please see Figure 1

Comment 9. Methods e. Screening of citations usually involves two stages - some citations can be filtered out based on the title and abstract alone, while others require a reading of the full text. This should be mentioned and included in Figure 1.

Response 9. Done

Please see Figure 1

Comment 10. Methods f. In Figure 1, no explanation for the superscript ** has been provided.

Response 10. We detail this point.

Please see Figure 1

Comment 11. Methods g. Figure 1 refers to 123 citations being retrieved from "registers" - which registers were searched? This should be mentioned in the text as well.

Response 11. We changed the PRISMA flow diagram to a more suitable model.

Comment 12. Methods h. The authors refer to "automation tools" being used to mark records as ineligible in Figure 1. Which tool(s) were used?

Response 12.

Change in the manuscript:

Search results were uploaded into EndNote X9 for a first screening, subsequently those files were exported to Covidence for a formal screening.

Comment 13. Methods i. The authors have used the Newcastle-Ottawa Scale (NOS) to assess study quality. Is this scale appropriate for evaluating the quality of gene expression studies, or are there more specific tools available?

Response 13. We improve the quality assessment.

Change in the manuscript:

Furthermore, all studies were critically appraised using ROBINS-I tool according to: intervention bias, missing data, confounding factors, outcome bias, report, selection and overall risk bias.

Please see Figure 2

Comment 14. Methods j. What was the rationale for using a cut-off of 6 stars or more on the NOS for including studies in the review? The developers of the tool themselves acknowledge that there is no clear evidence currently available to support such a cut-off.

Response 14. You are right. No cut-off was considered.

Change in the manuscript:

2.4. Quality assessment

The Newcastle–Ottawa Scale (NOS) was applied to assess the quality of the eligible articles. NOS involves three perspectives: study group selection, group comparability and whether the exposure or the outcome of interest for a case–control study is listed in the scale. Each study can receive a maximum of nine stars.

Comment 15. Methods h. How were confounding factors (e.g., victims' age, gender, or the presence of a medical or psychiatric diagnosis) taken into consideration when appraising the results of included studies?

Response 15. According to our primary aim, for this time our purpose just was to detailed and pooled data. We extracted such confounding factors (psychiatric diagnosis, mean and range age and the gender) of the patients who died by suicide; see Table 2.

Comment 15. Results a. The numbers provided in Section 3.1 do not match those provided in Figure 1; please ensure that these details are consistent throughout the paper.

Response 15. Checked

Comment 16. Results b. It is not clear what is meant by "most frequent" and "most recurrent" in section 3.2; the two terms are practically synonymous, but different brain regions are listed in each case.

Response 16. This section was re-written

Change in the manuscript:

Concerning, the regions and structures of the brain analyzed for gene expression in suicide completers, the most frequent were: prefrontal cortex, anterior cingulate cortex, dorsolateral pre-frontal cortex, ventral prefrontal cortex, orbital frontal cortex, hypothalamus, amygdala and hippocampus

Comment 17. Results c. Figure 2 does not present the review results in a balanced manner; there is an undue focus on some results (GABA, glutamate, HPA axis, monoamines) while other such as CXCL and NOTCH genes have been ignored.

Response 17. Thanks. We follow your suggestion

Comment 18. Results d. Was any analysis of the relationship between study quality and results attempted? If not, why?

Response 18. Thanks, we clarified some quality outcomes.

Change in the manuscript:

In the other hand, when the quality measurements was applied we could observed that studies differ in methodological standardization (e.g. type of method applied in gene expression, percent of male/female, among others). Nevertheless, the quality level of the studies did not report an important evidence of bias. However, any results should be taken with caution.

Comment 19. Discussion a. This section should discuss methodological / quality-related concerns for the included studies, as well as the possible effect of confounders (see 2h. above) and not just repeat the study results.

Response 19. According to our aim, we recognized that we just detail and group the data of the gene expression studies. In consideration to your suggestion, we are preparing a meta-analysis.

Comment 20. Discussion b. As mentioned in 3c. above, the discussion should focus on all relevant / significant findings and not just the four to five groups of findings that have been repeated most often. Contemporary understanding of suicide has moved beyond simple models based on monoamine / amino acid transmitters to incorporate neural plasticity, stress response, gut-brain axis interactions, neuropeptides, etc. and any positive results pertaining to these (and other related) pathways should also be discussed.

Response 20. We added some extra information

Change in the manuscript:

Chemokines are small proteins with several implications in neuroendocrine regulation, blood barrier permeability control, pre- and post-synaptic modulation among other essential activities for the normal functioning of the relation between central nervous system and immune system(Nakagawa & Chiba, 2015; Stuart & Baune, 2014). For that reason is not surprisingly, that chemokines such as CXCL1, CCL2 among others have been implicated in a number of neurological diseases. In fact, some paper suggest that the disruption of the chemokines functions previously mentioned in neurodevelopmental periods or in later life contribute to the pathophysiology of psychiatric traits (e.g. suicide)(Nakagawa & Chiba, 2015). Namely, CXCL proteins are involved in the inhibition of glutamatergic activity in hippocampal neurons and regulation processes of neuroplasticity (Rogers et al., 2011; Tokac et al., 2016).

Reviewer 2 Report

The authors analyzed the expression of genes involved in neurotransmission, neuroinflammation and neuro-regeneration processes in different brain areas of suicide completers and non-completers. The analyzed data came from number of publications of different authors, laboratories and countries. The collected data have been ordered in three tables, which are helpful to overview by readers, though not all abbreviations used have been explained.

The review article has some value, however, the analysis and discussion need to be performed more profoundly to enrich conclusions, which are not satisfactory and rather poor.  Moreover, English requires correction.

English needs correction.

Author Response

REVIEWER 2

Comment 1. The authors analyzed the expression of genes involved in neurotransmission, neuroinflammation and neuro-regeneration processes in different brain areas of suicide completers and non-completers. The analyzed data came from number of publications of different authors, laboratories and countries. The collected data have been ordered in three tables, which are helpful to overview by readers, though not all abbreviations used have been explained.

Response 1. Thanks. This was performed.

Comment 2. The review article has some value, however, the analysis and discussion need to be performed more profoundly to enrich conclusions, which are not satisfactory and rather poor. Moreover, English requires correction.

Response: We appreciate the valuable comments of the reviewer #2. We have revised and rewrite the abstract, introduction and discussion section. Also, we apologize for the mistakes in the manuscript. We also carefully checked and corrected the entire manuscript for grammatical errors.

Reviewer 3 Report

A very nice review and a good perspective as this will attract the readers of the journal and further interest (e.g. citations etc). Graphics and tables are very good.  Good job to the authors

Author Response

REVIEWER 3

Comment 1. A very nice review and a good perspective as this will attract the readers of the journal and further interest (e.g. citations etc). Graphics and tables are very good.  Good job to the authors

Response 2. Thanks.

Reviewer 4 Report

-  I have major misgivings regarding "Table 1" - rows should be grouped so that it makes sense. Either by country or by Brodmann's area... Further in that same table, abbreviations in the column " Brain region/structure..." have to be explained in full - use a footnote. Also, I think the audience would favor the year when a certain study was published.- Another issue - references are mostly 10 years old - please, reconsider this. And feel free to add a few references, since systematic reviews should have maybe RELEVANT references.

none

Author Response

REVIEWER 4

Comment 1. I have major misgivings regarding "Table 1" - rows should be grouped so that it makes sense. Either by country or by Brodmann's area... Further in that same table, abbreviations in the column " Brain region/structure..." have to be explained in full - use a footnote. Also, I think the audience would favor the year when a certain study was published.

Response 1. We follow your suggestion in Table 1

Comment 2. Another issue - references are mostly 10 years old - please, reconsider this. And feel free to add a few references, since systematic reviews should have maybe RELEVANT references.

Response: We appreciate the valuable comment of the reviewer #4. We have revised the manuscript by adding some relevant references. However, we keep classic or historical references.  

Round 2

Reviewer 2 Report

The authors have improved the manuscript, but English still needs correction .

English still needs careful correction (e.g. Figure 3 "Alterations Systems")